## Resource

# High-throughput identification of RNA nuclear enrichment sequences

Chinmay J Shukla[1,2,3,4], Alexandra L McCorkindale[1,5], Chiara Gerhardinger[1,2], Keegan D Korthauer[3,6], Moran N Cabili[2], David M Shechner[1,2], Rafael A Irizarry[3,6], Philipp G Maass[1,‡] & John L Rinn[1,2,7,†,‡,*]

## Abstract

In the post-genomic era, thousands of putative noncoding regulatory regions have been identified, such as enhancers, promoters, long noncoding RNAs (lncRNAs), and a cadre of small peptides. These ever-growing catalogs require high-throughput assays to test their functionality at scale. Massively parallel reporter assays have greatly enhanced the understanding of noncoding DNA elements *en masse*. Here, we present a massively parallel RNA assay (MPRNA) that can assay 10,000 or more RNA segments for RNA-based functionality. We applied MPRNA to identify RNA-based nuclear localization domains harbored in lncRNAs. We examined a pool of 11,969 oligos densely tiling 38 human lncRNAs that were fused to a cytosolic transcript. After cell fractionation and barcode sequencing, we identified 109 unique RNA regions that significantly enriched this cytosolic transcript in the nucleus including a cytosine-rich motif. These nuclear enrichment sequences are highly conserved and over-represented in global nuclear fractionation sequencing. Importantly, many of these regions were independently validated by single-molecule RNA fluorescence *in situ* hybridization. Overall, we demonstrate the utility of MPRNA for future investigation of RNA-based functionalities.

**Keywords** *de novo* inference of regions; high-throughput reporter assay; lncRNA; nuclear localization; RNA

**Subject Categories** Methods & Resources; RNA Biology; Systems & Computational Biology

**The EMBO Journal (2018) 37: e98452**

See also: **F Agostini** *et al* (March 2018)

## Introduction

One of the biggest surprises since the sequencing of the human genome has been the discovery of thousands of functional noncoding regions (Rinn & Chang, 2012; Hon *et al*, 2017). This includes new DNA enhancer elements, promoters, small RNAs, long noncoding RNAs (lncRNAs), and small peptides (20–70 amino acids) that are encoded in regions previously annotated as lncRNAs. Underscoring the importance of these elements are their disease associations and functional roles in the regulation of transcription (Jin *et al*, 2011; Morris & Mattick, 2014; Anderson *et al*, 2015; Gong *et al*, 2015; Hon *et al*, 2017). The ever-growing collection of noncoding annotations has motivated technological advances to characterize these elements and assay for their functional roles in a high-throughput manner. For example, the capacity to synthesize pools comprised of more than 100,000 individual DNA oligos has led to massively parallel reporter assays (MPRA) that have been applied to identify noncoding DNA elements, such as enhancers and promoters, on a genome-wide scale (Patwardhan *et al*, 2009; Melnikov *et al*, 2012; Oikonomou *et al*, 2014; Rosenberg *et al*, 2015; Ernst *et al*, 2016). These studies have turbo-boosted our understanding of functional DNA elements and their upstream regulatory factors. Addressing RNA functionalities in a similar manner has many challenges, remains limited, and is poorly scalable. Yet, such an assay would hold great promise to understand fundamental aspects of lncRNA biology through the identification of functional sequences and structures.

Central to RNA-based functionality is subcellular localization, which influences the biogenesis and function of mRNAs and lncRNAs alike. RNA localization provides a fundamental mechanism through which cells modulate and utilize the functions encoded in their transcriptomes (Davis & Ish-Horowicz, 1991; Bullock & Ish-Horowicz, 2001; Johnstone & Lasko, 2001; Lin & Holt, 2007; Paquin & Chartrand, 2008; Martin & Ephrussi, 2009; Zhang *et al*, 2014; Hacisuleyman *et al*, 2016). This spatial layer of post-transcriptional

1   Department of Stem Cell and Regenerative Biology, Harvard University, Cambridge, MA, USA
2   Broad Institute of MIT and Harvard, Cambridge, MA, USA
3   Department of Biostatistics and Computational Biology, Dana-Farber Cancer Institute, Boston, MA, USA
4   Program in Biological and Biomedical Sciences, Harvard Medical School, Boston, MA, USA
5   Berlin Institute for Medical Systems Biology, Max Delbrück Center for Molecular Medicine, Berlin, Germany
6   Department of Biostatistics, Harvard T.H. Chan School of Public Health, Boston, MA, USA
7   Department of Pathology, Beth Israel Deaconess Medical Center, Boston, MA, USA
    *Corresponding author. Tel: +1 303 735 7218; E-mail: john.rinn@colorado.edu
    ‡ These authors contributed equally to this work
    †Present address: Department of Biochemistry, University of Colorado BioFrontiers, Boulder, CO, USA

gene regulation is known to be critical in a variety of contexts, including asymmetric cell divisions (Paquin & Chartrand, 2008), embryonic development (Davis & Ish-Horowicz, 1991; Bullock & Ish-Horowicz, 2001; Johnstone & Lasko, 2001), and signal transduction (Lin & Holt, 2007). Previous work has identified a small number of *cis*-acting mRNA localization elements, using genetic approaches or hybrid reporter constructs to decipher sequences required for localization to specific parts of the cell (Bullock & Ish-Horowicz, 2001; Martin & Ephrussi, 2009). These elements are often located in 3′ untranslated regions and range from five to several hundred nucleotides in length (Bullock & Ish-Horowicz, 2001; Miyagawa *et al*, 2012; Zhang *et al*, 2014; Hacisuleyman *et al*, 2016). Yet, the sequences and structures responsible for RNA localization remain inchoate.

In contrast to mRNAs—which are exclusively cytosolic—most lncRNAs are predominantly enriched in the nucleus (Derrien *et al*, 2012). Consistent with their localization patterns, several examples of lncRNAs (*XIST* (Brown *et al*, 1992; Lee & Bartolomei, 2013), *FIRRE* (Hacisuleyman *et al*, 2016), *MALAT1* (Gutschner *et al*, 2013), *NEAT1* (Clemson *et al*, 2009), *PVT1* (Tseng *et al*, 2014), *GAS5* (Kino *et al*, 2010), *PINT* (Marín-Béjar *et al*, 2013), and many others) perform key nuclear roles during development and are believed to be crucial in nuclear organization (Rinn & Guttman, 2014). This is surprising since mRNAs and lncRNAs share similar biogenesis and post-transcriptional features (Cabili *et al*, 2011; Ni *et al*, 2013; Guttman & Rinn, 2012; e.g., m7-G cap and polyA tail), which usually trigger RNA export to the cytosol. This raises a more general question: Is there a universal nuclear localization motif harbored within lncRNAs (Zhang *et al*, 2014), or is nuclear localization imparted by larger RNA domains specific to individual transcripts (Hacisuleyman *et al*, 2016)? Addressing this question requires a high-throughput assay that can screen for RNA-based functionalities.

Toward this goal, we have developed and optimized such a massively parallel RNA assay (MPRNA). Briefly, we developed a construct that expressed and appends thousands of 110mer RNA sequences—each uniquely barcoded—to a cytosolic-localized reporter transcript: a noncoding, frame-shifted variant of *Sox2*, which we hereafter refer to as *fsSox2* (see Materials and Methods). By sequencing barcodes in nuclear fractions versus the total barcode population, we can simultaneously assess each 110mer that was sufficient to retain *fsSox2* in the nucleus. To control for the possibility that sequences larger than 110 nucleotides (nt) might be required for nuclear retention, we designed a densely overlapping pool of oligos so that, on average, every unique 10 nt are independently assayed. This was optimized to develop a robust statistical method that leverages the interdependencies and variances of each 110mer to identify larger RNA domains enriched in the nucleus.

As a first application, we performed MPRNA across 38 lncRNAs with varying degrees of subcellular localization patterns as previously determined by single-molecule RNA fluorescence *in situ* hybridization (smFISH; Cabili *et al*, 2015). We identified 109 unique nuclear enrichment sequences derived from 29 of the 38 lncRNAs tested, including the known RNA localization regions for *MALAT1* (Miyagawa *et al*, 2012). Interestingly, a global motif analysis of these regions uncovered a cytosine-rich (C-rich) motif that is over-represented in many of the nuclear enrichment regions. Consistent with a possible global role of the C-rich motif for localization, these regions tend to be more conserved and are generally

nuclear-enriched in global nuclear versus cytoplasmic RNA sequencing (RNA-seq) experiments from the ENCODE consortium. Notably, a very similar motif was also identified in an independent study (Lubelsky & Ulitsky, 2018). Finally, we independently validated the capability of these domains to impart nuclear localization by smFISH of *fsSox2* appended with the putative nuclear enrichment sequences identified by our MPRNA. Collectively, we demonstrate that the MPRNA methodology could be universally applicable to identify active RNA elements sufficient for any cellular process that can be physically and functionally separated.

# Results

## Design and optimization of a massively parallel RNA assay (MPRNA)

In order to identify RNA sequences that drive lncRNA nuclear enrichment, we developed a high-throughput approach for identifying nuclear enrichment elements. First, we designed a pool of 11,969 153-nt oligos representing 38 lncRNAs with diverse subcellular localization patterns: from single nuclear foci (*e.g., XIST, ANRIL, ANCR, PVT1, KCNQ1OT1, FIRRE*) to broadly diffuse cytosolic patterns (*e.g., NR_024412, NR 033770*; Cabili *et al*, 2015). We designed the oligo-pool to tile each of the 38 lncRNAs with a 10-nt shift between sequential oligonucleotides. This densely overlapping tiling approach offers us a unique advantage of allowing the computational "stitching" of sequential oligos (Jaffe *et al*, 2012; preprint: Korthauer *et al*, 2017), thus enabling identification of longer regions required for nuclear enrichment. Second, we cloned the pool of oligonucleotides to the 3′ end of a cytosolic-localized *Sox2* construct (*fsSox2*). As previously shown (Haciuselyman *et al*, 2016), we used *fsSox2* instead of regular *Sox2* to avoid any unwanted translation artifacts. The oligo-pool was expressed in HeLa cells, followed by subcellular fractionation, and targeted RNA-seq of unique barcodes to determine the enrichment of each *fsSox2* variant in the nucleus relative to the total barcode representation in total RNA (Fig 1A, Table EV1, Materials and Methods). The assay was performed as six biological replicates to ensure sufficient statistical power for our analytical model, and accurately estimate in-group variance (see below, Materials and Methods).

We ranked candidate localization regions using a newly defined summary statistic that generates a null distribution by permuting sample labels, which is used to assign *P*-values (Fig 1B–D; Materials and Methods). Our approach overcomes the inter-replicate variability inherent in high-throughput reporter assays and allows us to sensitively and accurately discover nuclear-enriched RNA segments spanning up to hundreds of base pairs, which we term "differential regions" (DRs).

At each stage of the MPRNA, we used quality controls (Fig EV1), and to prove the principle of our assay and analytical method, we first focused on a well-established nuclear lncRNA, *MALAT1*. Previous work demonstrated that two elements within *MALAT1* ("Region E" and "Region M") act as potent nuclear localization signals (Miyagawa *et al*, 2012). We examined the nuclear enrichment of all *fsSox2* pool variants bearing oligos derived from *MALAT1* (Materials and Methods). Consistent with the previous study, nucleotides derived from Region E and Region M were highly

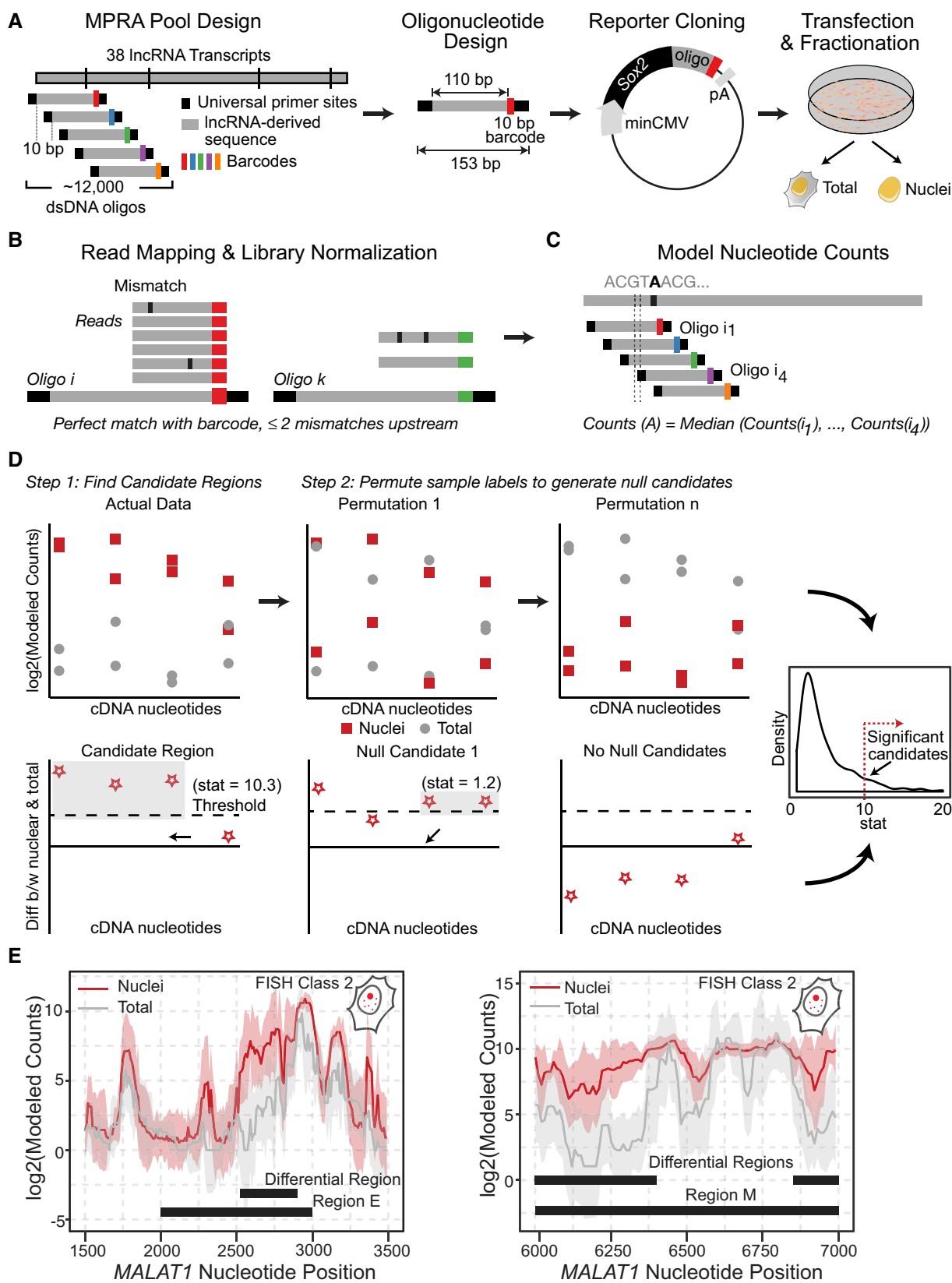

**Figure 1.**

**Figure 1.   A Massively Parallel RNA Assay (MPRNA) to identify RNA nuclear enrichment signals.**

A   Experimental overview. *Far left*: oligonucleotide pool design. Double-stranded DNA (dsDNA) oligonucleotides were designed by computationally scanning 38 parental lncRNA transcripts (Table EV1) in 110-nt windows, with 10-nt spacing between sequential oligos. These lncRNA-derived sequences (gray) were appended with unique barcodes and universal primer binding sites, resulting in a pool of 11,969 oligos of 153 bp (Table EV1). The vertical lines in the lncRNA denote splice junctions. *Second from left*: schematic summarizing the design of each oligonucleotide. *Second from right*: reporter design. The oligonucleotide pool was cloned into a reporter plasmid as fusion transcript 3′ of *fsSox2* (minCMV, minimal CMV promoter; pA, polyadenylation sequence). *Far right*: MPRA workflow. The *fsSox2*~oligo reporter pool was transiently transfected into HeLa cells. Following 48 h of expression, cells were harvested and fractionated to isolate nuclei, and the nuclear enrichment of each oligo was quantified by targeted RNA sequencing. Matched whole-cell lysates from unfractionated cells served as controls.

B   Read mapping and normalization. A perfect match between the first 10 nt of the read and the barcode sequence was used to "map" the read. To guarantee robustness of the mapping procedure, we allowed for no more than two mismatches within the 90 basepairs upstream of the barcode (see "mapReads" function in our analysis package—please refer to the Code availability section). Counts were normalized for library size using the "normCounts" table (see analysis package and GEO data—please refer to the Code and Data availability sections).

C   Counts for each nucleotide were modeled based on the normalized counts for each oligo. When nucleotide "A" overlapped with oligos $i_1$, $i_2$, $i_3$, and $i_4$, counts for this nucleotide were modeled by the median of counts for each of the individual oligos ($i_1$–$i_4$; see "modelNucCounts" function in our analysis pipeline).

D   The nucleotide counts were then used to infer differential regions by (1): finding candidate regions and assigning a summary statistic to each one of them and next (2): generating null candidates by permuting sample labels and using them to assign an empirical *P*-value to our candidate regions from step 1 to identify significant regions.

E   Differential region-calling correctly identifies nuclear retention elements in *MALAT*. Solid lines: per-nucleotide abundances in the nuclear (red) and whole-cell (gray) fractions, modeled for each nucleotide position along the *MALAT1* transcript, based on the aggregate behavior of all oligos containing that nucleotide (shaded regions: ±SD, medians of six biological replicates).

enriched in the nucleus, compared to those from elsewhere in *MALAT1*. This finding demonstrates that our assay can recapitulate known RNA localization signals and that our analytical approach can identify localization domains longer than 110 nt (Fig 1E).

**MPRNA-based identification of RNA nuclear enrichment regions**

Next, we sought to agnostically and systematically investigate nuclear enrichment regions harbored within the selected 38 lncRNAs. Our analysis identified 109 DRs (FDR < 0.1) originating from 29 distinct lncRNAs that were significantly enriched in nuclear fractions relative to whole-cell lysates (Table EV2). Two of these DRs overlap and subsume the *MALAT1* Region M while another overlap with Region E (Fig 1E). To confirm that our approach was robust, we compared the significant DRs to all other regions represented in the pool and found them significantly more nuclear-enriched ($P < 1/10^6$, Mann–Whitney test; Materials and Methods). The localization patterns of the selected 38 lncRNAs have been previously parsed into five smFISH classes (Cabili *et al*, 2015). These included lncRNAs ranging from strictly nuclear (Class I) to cytoplasmic (Class V), with three intermediate classes (classes II–IV). The MPRNA discovered DRs derived from lncRNAs in all five FISH classes (Fig 2A–E). To compare DRs found across different FISH classes, we normalized for the length of transcripts tiled across each FISH class. After normalization, we found that the number of DRs per kb was broadly similar within each FISH class (Fig 2F). Interestingly, many Class I lncRNAs harbor multiple DRs, possibly indicating the presence of a redundant nuclear localization motif. For example, we discovered 18 DRs in *XIST* and 10 DRs in *MALAT1* and some of the DRs we discovered in *XIST* overlap with the previously described *XIST* repeat elements RepC and RepD (Appendix Fig S1A; Brown *et al*, 1992). By contrast, we only discovered 1 DR in predominantly cytosolic lncRNAs such as *NR_023915* and *NR_040001*. Interestingly, while 60% of the lncRNAs in the pool were nuclear, 66% of the lncRNAs lacking DRs were predominantly cytosolic.

We further analyzed the evolutionary conservation, length distribution, and sequence content of DRs for putative nuclear localization sequences. We used phastCons (Siepel *et al*, 2005, 2006) scores to assess evolutionary conservation, and we observed significantly

higher scores among our DRs than in other lncRNA regions tiled by our MPRNA (Fig 2G; $P < 1/10^6$, Mann–Whitney test; Materials and Methods). The lengths of the DRs ranged from 80 to 740 nt, with an average of 300 nt (Appendix Fig S1B). While we detected a weak correlation between the length of a given lncRNA and number of DRs within (Appendix Fig S1C), this analysis is confounded by the different length of lncRNAs across the five FISH classes. Finally, we did not observe a difference in GC content between the DRs and other sequences within the tiled lncRNAs (Appendix Fig S1D).

We hypothesized that the identified DRs might harbor common sequence motifs or preferences. To test this, we searched for motifs that were more prevalent among the DRs than in other regions of the lncRNAs, using the MEME software package (Machanick & Bailey, 2011). We identified a 57-nt motif (*E*-value = 3.7e-10) occurring 18 times exclusively in *XIST* but not elsewhere in the human genome (Fig 3A–C). Another 15-nt C-rich motif (*E*-value = 9.0e-10) was found in 52 DRs of 21 different lncRNAs (Fig 3D–F), and we discovered four additional motifs closely related to the ones described here (Appendix Fig S2A–D). Similarly, *k*-mer analysis (Le Cessie & Van Houwelingen, 1992) revealed several C-rich 4-mers that were mildly predictive of a DR (Appendix Fig S2E). In total, we discovered six motifs and confirmed that the nucleotides overlapping these motifs were significantly enriched in the nucleus ($P < 1/10^6$, Mann–Whitney test, Materials and Methods), compared to all other regions tiled in our MPRNA (Fig 3G). Since the C-rich motif occurred in many distinct DRs of diverse lncRNAs, we postulated that this motif could function as a global RNA nuclear localization element. To test this, we examined the nuclear versus cytoplasmic localization of both human lncRNA and mRNA transcripts containing this motif, from the ENCODE consortium fractionation RNA-seq data (ENCODE Project Consortium, 2012). For both lncRNAs and mRNAs, we observed a modest-yet-significant increase ($P < 1/10^6$, Mann–Whitney test) in nuclear localization of transcripts containing the C-rich motif across all 11 ENCODE TIER 2 cell lines (Fig 3H and I, Appendix Fig S3). Interestingly, we note that while the effect of the motifs was significant for both lncRNAs and mRNAs, the effect size was larger in motif-harboring lncRNAs. Collectively, these results demonstrate the power of our MPRNA to discover potential functional elements that may be missed by classic RNA localization studies.

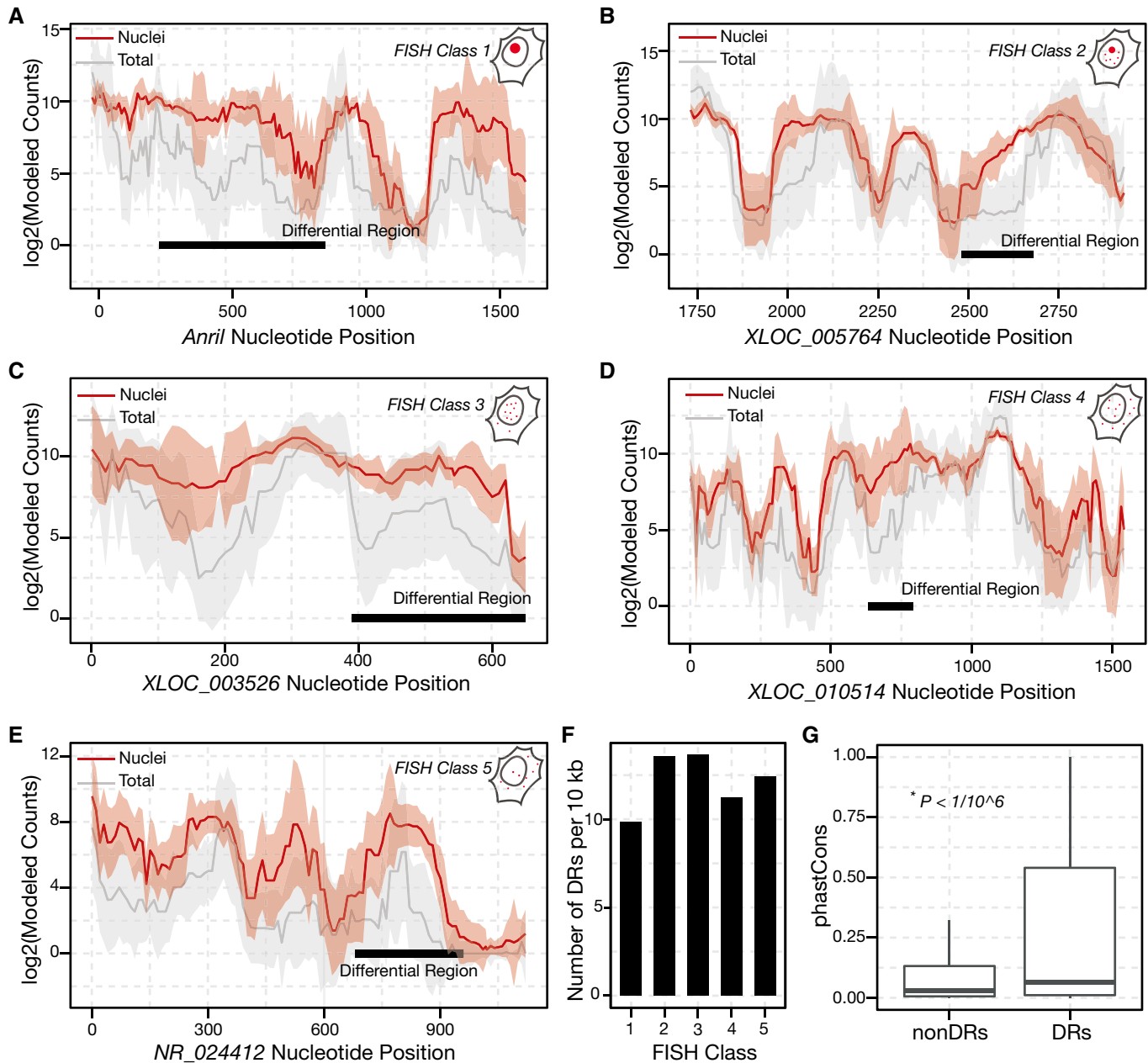

**Figure 2.    Novel lncRNA nuclear enrichment signals.**

A–E    Identification of differential regions (DRs) within lncRNAs with different subcellular localization patterns. Data are depicted as in Fig 1E. Established subcellular localization patterns range from (A), occupying a single, prominent nuclear focus (*ANRIL*, FISH Class 1), to (E), exhibiting a diffuse, mostly cytosolic pattern (*NR_024412*, FISH Class 5; Cabili *et al*, 2015).

F    The number of DRs discovered per 10 kb of lncRNA sequence tiled is similar for each FISH Class.

G    DRs are more conserved than most lncRNA sequences. Boxplot of phastCons scores comparing nucleotides within DRs (*red*), to all other nucleotides within the oligo-pool (*gray*). *P*-value: Mann–Whitney Test. The solid horizontal line is the median while the lower and upper hinges correspond to the first and third quartiles (the 25th and 75th percentiles). The upper whisker extends from the hinge to the largest value no further than 1.5 × IQR from the hinge (where IQR is the inter-quartile range). The lower whisker extends from the hinge to the smallest value at most 1.5 × IQR of the hinge. Data beyond the end of the whiskers are outliers and are plotted individually.

## Single-molecule RNA-FISH validation of nuclear enrichment motifs and domains

We independently tested if the motifs identified by our MPRNA are sufficient for nuclear localization using a smFISH-based reporter assay (Fig 4A). Briefly, we appended consensus motif sequences (small motifs and long DRs) to the 3′ end of the cytosolic *fsSox2* reporter and electroporated these constructs into HeLa cells (Hacisuleyman *et al*, 2016; Fig 4B and C). We then performed smFISH (Levesque & Raj, 2013) using RNA probes antisense to

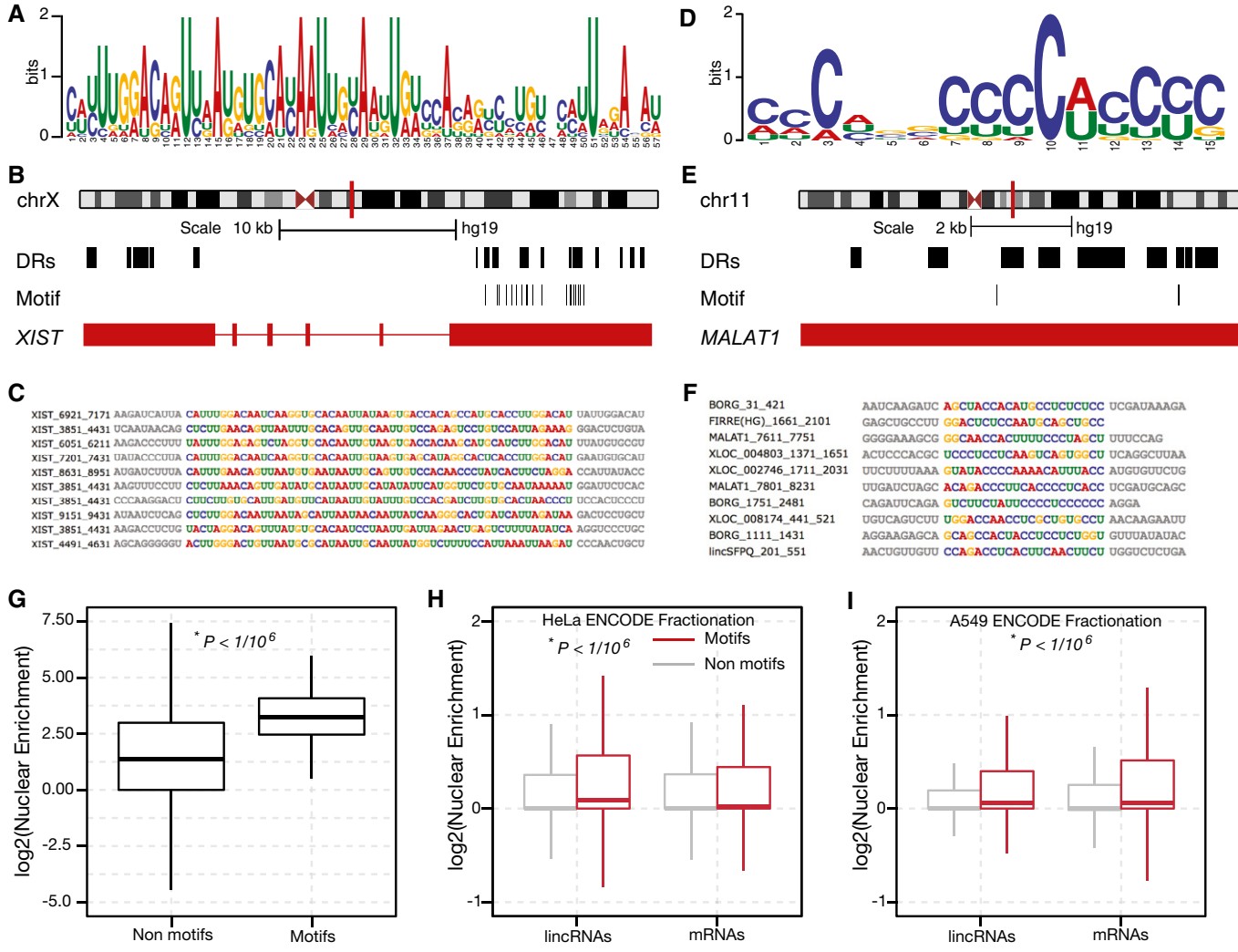

**Figure 3.  Motifs enriched in lncRNA nuclear enrichment signals.**

A   Position weight matrix (PWM) for a novel 57-nt motif enriched within the DRs of *XIST* (*E*-value < 0.05).
B   Occurrences of this motif throughout the *XIST* locus.
C   Multiple sequence alignments of the incidences of the *XIST* motif (*colored nucleotides*) within the *XIST* DRs. Adjoining sequences are colored in gray.
D   PWM for a novel C-rich 15-nt motif enriched within the DRs of 21 different lncRNAs (*E*-value < 0.05).
E   The occurrences of this motif throughout the *MALAT1* locus.
F   Multiple sequence alignments of different instances of this motif (*colored nucleotides*), as they appear in the DRs of the indicated lncRNAs.
G   Oligos bearing the novel motifs described in Fig 2A–F and Appendix Fig S2 are significantly enriched in nuclear fractions, relative to all other oligos in the MPRNA. *P*-value: Mann–Whitney test. The solid horizontal line is the median while the lower and upper hinges correspond to the first and third quartiles (the 25th and 75th percentiles). The upper whisker extends from the hinge to the largest value no further than 1.5 × IQR from the hinge. The lower whisker extends from the hinge to the smallest value at most 1.5 × IQR of the hinge. Data beyond the end of the whiskers are outliers and are plotted individually.
H, I   Novel nuclear enrichment motifs influence the localization of endogenous human transcripts. Comparison of the nuclear enrichment of both human mRNA and lncRNA transcripts with at least one occurrence of the discovered motifs, relative to all other transcripts, in HeLa and A549 cells (ENCODE Project Consortium, 2012). *P*-value: Mann–Whitney test. The solid horizontal line is the median while the lower and upper hinges correspond to the first and third quartiles (the 25th and 75th percentiles). The upper whisker extends from the hinge to the largest value no further than 1.5 × IQR from the hinge. The lower whisker extends from the hinge to the smallest value at most 1.5 × IQR of the hinge. Data beyond the end of the whiskers are outliers and are plotted individually.

*fsSox2*, followed by double-blinded spot quantification using Star-Search (Levesque & Raj, 2013; Materials and Methods). We observed that ~30% of *fsSox2*-only transcripts were detected in the nucleus, and appending the repetitive *XIST* motif increased nuclear localization to ~40% (Fig 4D; *P* = 0.03, Mann–Whitney test).

Appending the C-rich motif did not significantly affect the localization of *fsSox2* (Fig 4D).

We next investigated whether the longer DRs identified by our MPRNA might impart a stronger effect on nuclear localization. Therefore, we expressed *fsSox2*-DR fusion transcripts (DRs from *MALAT1*,

625 nt*; TUG1, 721 nt; and XIST, 581 nt)* in HeLa cells, and compared their subcellular localization to that of the *fsSox2*-only transcript by smFISH. As expected, we found that the *MALAT1* "Region M" significantly increased nuclear enrichment of *fsSox2* (Fig 4D; $P < 1/10^6$, Mann–Whitney test). Similarly, the *TUG1* DR and *XIST* DR (which harbors the *XIST* motif) also promoted nuclear enrichment of *fsSox2* (Fig 4D; $P < 1/10^6$, Mann–Whitney test; Materials and Methods). Thus, the longer DRs identified in our MPRNA are sufficient to affect the nuclear enrichment of an otherwise-cytosolic transcript.

## Discussion

Here we present a methodology to systematically assay RNA-based functionalities in an unbiased manner. As a first application of MPRNA, we simultaneously interrogated over 10,000 RNA sequences for their ability to impart changes in subcellular localization. We found that our pool design strategy also allows us to leverage redundancy, variance interdependencies, and statistical Materials and Methods to identify larger RNA regions that provide signal in MPRNA.

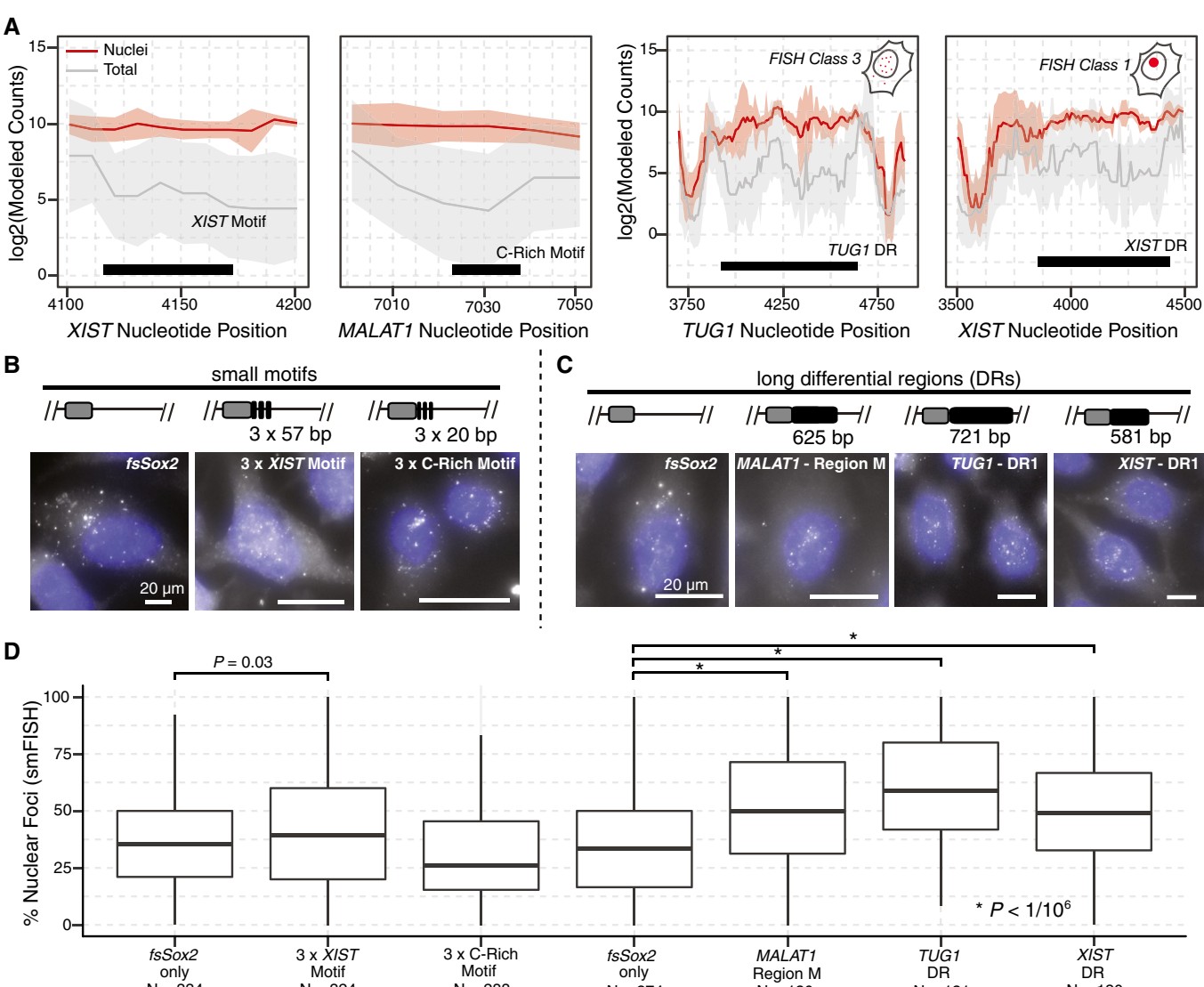

**Figure 4. Differential regions are sufficient to redirect RNA subcellular localization.**

A Representative *XIST* and C-rich motif regions and novel differential regions (DRs) from lncRNAs *TUG1* and *XIST* that are examined in (B–D). Data depicted as in Fig 1E.

B Experimental overview and examples of smFISH experiments. *fsSox2* reporter constructs were fused to individual small motifs and long DRs. Representative smRNA-FISH images: (*left*) unmodified *fsSox2* reporter, (*middle*) *fsSox2* fused to three tandem *XIST* motifs, and (*right*) *fsSox2* fused to three tandem instances of the C-rich motif (scale bars = 20 μm, blue: Hoechst 33342).

C Representative smRNA-FISH images of (*left*) unmodified *fsSox2*, *MALAT1* Region M (*second from left*), *TUG1* DR (*second from right*), and *XIST* DR (*right*).

D smFISH quantification of the nuclear localization of *fsSox2* reporter constructs fused to the indicated motifs and DRs (*P*-value: Mann–Whitney test). The solid horizontal line is the median while the lower and upper hinges correspond to the first and third quartiles (the 25th and 75th percentiles). The upper whisker extends from the hinge to the largest value no further than 1.5 × IQR from the hinge. The lower whisker extends from the hinge to the smallest value at most 1.5 × IQR of the hinge. Data beyond the end of the whiskers are outliers and are plotted individually.

We applied MPRNA to nuclear localization and demonstrated that we can identify active RNA regions and glean insights into RNA biology. For example, across the 38 lncRNAs tested, we find many regions greater than 300 nts (median DR length was 290 nts) as the drivers of nuclear enrichment. This includes known regions required to retain *MALAT1* that was recapitulated in the MPRNA, as well as identification of several novel regions. Consistently, independent smFISH analysis confirmed the sufficiency of these regions to promote *fsSox2* localization to the nucleus. Moreover, we observed that lncRNAs that are more nuclear tend to contain a greater number of regions flagged as DRs by our assay. Together, these independent results converge on the reproducibility and robustness of the MPRNA.

Deeper analysis of potential common sequence motifs underlying these longer RNA nuclear enrichment regions uncovered a C-rich sequence motif that is over-represented in these regions. Supporting this finding, C-rich motifs are modestly enriched in total RNA sequencing of nuclear versus cytoplasmic samples provided by ENCODE. Also, a similar motif was identified in an independent study using a similar approach (Lubelsky & Ulitsky, 2018). In contrast, we were not able to validate the nuclear localization properties of this sequence by smFISH. This could be due to the C-rich motif requiring a larger "linker" region or a particular stoichiometry for RNA secondary structure. Deeper investigation of these possibilities is among many other directions for future mechanistic work.

Interestingly, *XIST* also exhibited a small 57-nt motif that was repeated 18 times within the mature transcript, but is not detectable elsewhere in the genome. Remarkably, the *XIST* motif was sufficient —albeit modestly—to enrich *fsSox2* in the nucleus as determined by smFISH. This, combined with our finding that longer regions are sufficient to alter the localization of a cytoplasmic transcript, further suggests that smaller motifs could be important for nuclear localization but likely require a larger RNA context.

Together, these findings from this first application of MPRNA raise several new hypotheses for future experimental investigation. For example, what are the constituent proteins that bind to the nuclear enrichment sequences? Considering that our initial results suggest that longer RNA sequences are more effective in nuclear enrichment, it is likely that several proteins could scaffold a given region for nuclear retention. With our initial map of 109 regions, we can now hone in on these as an initial test for relevant RNA–protein interactions through various additional experimental approaches.

One advantage to the MPRNA strategy is that it is a universally applicable logical framework to understand RNA biology at a global level. More focused studies such as determining structural features that drive specific RNA–protein interactions could also benefit from a MPRNA approach. By designing an oligo-pool containing numerous sequence variants and compensatory mutations for specific RNA–protein interaction sites, one could gain insights into lncRNA structure–function relationships. For example, by combining RNA–protein binding assays (e.g., CLIP) with MPRNA, one could assay thousands of sequence and structural variants in parallel to determine common binding motifs or structures of RNA–protein interactions. We also envision developing additional MPRNA constructs to screen across diverse aspects of RNA biology, such as sequence requirements for splicing, gene regulation, or enhancer and suppressor activity.

Collectively, we have demonstrated that MPRNA is a robust and reproducible strategy to identify activity in RNA sequences. Notably, MPRNA can be applied to any assay where a separation of active versus inactive RNAs can be achieved. Overall, MPRNA can be combined with classic biochemical approaches to achieve the needed genome-scale to address many pressing RNA biology questions.

# Materials and Methods

### Oligo-pool design

We designed 153-mer oligonucleotides to contain, in order, the 16-nt universal primer site ACTGGCCGCTTCACTG, a 110-nt variable sequence, a 10-nt unique barcode sequence, and the 17-nt universal primer site AGATCGGAAGAGCGTCG. The unique barcodes were designed as described previously, while the variable sequences were obtained by tiling lncRNA sequences. The resulting oligonucleotide libraries were synthesized by Broad Technology Labs.

### ePCR amplification of oligo-pool

The synthesized oligo-pool was amplified by emulsion PCR (ePCR, Micellula DNA Emulsion & Purification Kit, Chimerx), according to the manufacturers' instructions. The ePCR primers were designed to add the AgeI/NotI restriction sites to the synthesized oligos for subsequent cloning (AgeI primer: AATAATACCGGTACTGGCC GCTTCACTG; NotI primer: GAGGCCGCG GCCGCCGACGCTCTTCC GATCT). To determine the oligos representation of the ePCR-amplified oligo-pool (based on the unique 3′ barcode of each oligo), 1 ng of the amplified oligo-pool was used as input for library preparation (see below) and sequenced on a MiSeq (SR, Illumina).

### Cloning

A minCMV promoter (5′-TAGGCGTGTACGGTGGGAGGCCTATAT AAGCAGAGCTCGTTTAGT GAACCGTCAGATCGC-3′) was cloned upstream of *fsSox2*. Similar to a previous publication (Hacisuleyman *et al*, 2016), we used a cytosolic-localized *fsSox2* reporter in order to avoid translation-derived artifacts. The ePCR-amplified oligo-pool and the identified motifs and candidate regions were digested with AgeI/NotI and inserted 3′ of *fsSox2*. For MPRNA cloning, the ligation reaction (100 ng backbone + 4× molar excess of oligo-pool) was transformed into 10× DH5α tubes (ThermoScientific). A total of 20 ampicillin LB plates were inoculated with the 10 transformation reactions and incubated overnight at 37°C. All bacterial colonies were then scraped in 5 ml of LB per plate and pooled, and the plasmids were purified with the endotoxin-free Qiagen Plasmid Plus Maxi kit (Qiagen). The cloned oligo-pool was then sequenced on the MiSeq to determine the oligo representation as described above.

### Cell fractionation

HeLa nuclear and cytoplasmic fractions were isolated as previously described (Hacisuleyman *et al*, 2016). The success of the

fractionations (Fig EV1B) was confirmed by qRT–PCR of the nuclear ncRNA NEAT1 and the cytoplasmic ncRNA SNHG5 in RNA isolated (see below) from whole cells, the pelleted nuclei, and from the cytoplasmic fractions.

### RNA extraction and qRT–PCR

RNA was isolated by TRIzol (ThermoScientific)—chloroform extraction, followed by isopropanol precipitation, according to standard procedures. 2 μg of BioAnalyzer-validated RNA was digested with recombinant DNase I (2.77 U/μl, Worthington #LS006353) at 37°C for 30 min, followed by heat inactivation at 75°C for 10 min. Reverse transcription was performed with SuperScript III cDNA synthesis kit (ThermoScientific). Quantitative RT–PCR was performed using the FastStart Universal SYBR Green Master mix (Roche) on an ABI 7900. Primers were as follows: NEAT1 forward TGATGCCACAACGCAGATTG, reverse GCAAACAGGTGGGTAGG TGA, and SNHG5 forward GTGGACGAGTAGCCAGTGAA, reverse GCCTCTATCAATGGGCAGACA. After processing the raw data by qPCR Miner (Zhao & Fernald, 2005), the efficiency of each primer set was used to calculate the relative initial concentration of each gene. The relative expression in the nuclear and cytoplasmic fractions was then calculated by normalization to that in the whole cell.

### Library preparation

Sequencing libraries were prepared by PCR amplification using PfuUltra II Fusion DNA polymerase (Agilent #600672) and primers designed to anneal to the universal primer site flanking the oligos and to add sequencing index barcode for multiplexing: forward caagcagaagacggcatacgagatCGTGATgtgactggagttcagacgtgtgctcttccgatct ACTGGCCGCTTCACTG, reverse AATGATACGGCGACCACCGAGAT CTACACTCTTTCCCTACACGACGCTCTTCCG ATCT (capital letters indicate (i) the index for the library, and (ii) the region complementary to the universal primer site). PCR amplification (initial denaturation 95°C—2 min; cycling 95°C—30 s, 55°C—30 s, 72°C—30 s; final extension 72°C—10 min) was carried out for 30 cycles followed by triple 0.6×, 1.6×, and 1× SPRI beads (Agencourt AMPure XP, Beckman Coulter) cleanup. The quality and molarity of the libraries was evaluated by BioAnalyzer, and the samples were sequenced in a pool of 6 on the Illumina HiSeq2500, full flow cell, single-read 100 bp. To ensure the transfection was successful, we required that at least 70% of the oligo-pool was represented back (i.e., had a count of at least one) in the sequencing sample (Fig EV1).

### Analyzing MPRNA data

#### Read mapping and obtaining counts table
To find a unique mapping location for the read, we ensured an exact match between the first 10 read nucleotides and a unique oligo barcode. To ensure that the correct oligo was identified using this barcode match, we allowed only two mismatches between the remaining 65 nts of the read sequence and the upstream oligo sequence corresponding to the unique barcode (Fig 1B). The resulting counts for each oligo in every sample (6 Nuclei and 6 Total) were compiled in a counts table (Fig 1B).

#### Normalizing the counts table
The counts table was normalized using a library size correction in order to facilitate comparing counts across samples with different sequencing depths. The library size was calculated as the total number of reads in each sample.

#### Modeling nucleotide counts from Oligo counts
The counts of a particular nucleotide were modeled by taking the median of counts for every oligo tiling the nucleotide (Fig 1C). Since the offset between subsequent oligos was usually 10 nucleotides, we obtained nucleotide counts also at a 10-nucleotide resolution. The resulting modeled nucleotide counts table (Table EV2) was used to infer differential regions.

#### Inferring differential regions from modeled nucleotide counts
There are two main steps in inferring differential regions from modeled nucleotide counts—(i) identifying potential candidate regions, and (ii) assigning a $P$-value for each potential candidate region (Fig 1D). We identified potential candidate regions by calculating the median of the difference between nuclear counts and total counts across all six replicates at each nucleotide and then grouping together neighboring points that exceeded a threshold, as described previously (Jaffe *et al*, 2012). We then defined a summary statistic for each region based on the differences between nuclear and total counts of each nucleotide in the region as well as the trend of these counts. To assess the uncertainty of this procedure, we generated a list of global null candidates by shuffling the sample labels and computed a summary statistic for these regions to form a null distribution. Then, we ranked each potential candidate region by comparing their respective summary statistic to the null distribution to obtain an empirical $P$-value. The $P$-values were converted to $q$-values using the Benjamini–Hochberg approach.

### Motif analysis

MEME (Machanick & Bailey, 2011) software package was used to find motifs enriched in differential regions. Specifically, we used the MEME function in the suite in the discriminative mode with DR sequences as the list of primary sequences and the other sequences in the pool as the controls. We ran MEME in different settings—OOPS and ANR—to ensure we found motifs that were repeating several times in a given DR and those only occurring once.

### *k*-mer enrichment

If sequence preferences are driven by more general sequence composition preferences that cannot be so easily represented by regular expression or position weight matrix motif models, then nuclear enrichment of DRs may be more effectively modeled by considering all $k$-mers. To this end, we performed a regression to assign weight coefficients to all $k$-mers for the DR sequences and non-DR sequences similar to the motif analysis using MEME as described previously. To avoid overfitting, we performed ridge regression (Le Cessie & Van Houwelingen, 1992), which minimizes not only the distance between model predictions and actual values but also the magnitude of the weights. We chose the alpha parameter that varies the emphasis of these two competing objectives by

evaluating fivefold cross-validated mean squared error over a parameter grid.

## Conservation analysis

The phastCons and phyloP scores (Siepel *et al*, 2005, 2006) for the whole genome were downloaded from UCSC genome browser. We extracted these scores for the DRs and shuffled control regions using a custom script. In order to account for natural conservation differences between lncRNAs and mRNAs as well as among different lncRNAs, the control regions were obtained by shuffling the DR sequences using shuffleBed but ensuring the new regions fell within exons of the lncRNAs the DRs were from. Finally, the scores were compared between DR and non-DR regions using the Mann–Whitney test.

## ENCODE fractionation RNA-Seq

We downloaded the raw RNA-Seq reads for the nucleus and cytosolic compartments from the ENCODE Project Consortium (2012) website. These reads were quantified using salmon (Patro *et al*, 2017) to obtain TPMs, and then, the nuclear/cytosolic TPMs of transcripts with the motif [found using the FIMO (Grant *et al*, 2011) software] were compared to all the other transcripts for both lncRNAs and mRNAs.

## Single-molecule RNA fluorescence *in situ* hybridization (smRNA-FISH)

Briefly, 70–80% confluent $1 \times 10^6$ HeLa (ATCC® CCL-2™) cells were electroporated with 2 μg of construct using the Amaxa® Cell Line Nucleofector® Kit R using program I-013, and cultured for 48 h in LabTek v1 glass chambers. smFISH was performed using Biosearch Technologies Stellaris® probes, as described previously (Hacisuleyman *et al*, 2016). RNA probes targeting and tiling the *fsSox2* exon were conjugated to Quasar 570. Nuclei were visualized with 4,6-diamidino-2-phenylindole (DAPI). Images were obtained using the Zeiss Cell Observer Live Cell microscope at the Harvard Center for Biological Imaging. For each field of view, at least 40 slices (each plane: 0.24 μm) were imaged, and z-stacks were merged with maximum intensity projections (MIP). *fsSox2* foci were computationally identified using the spot counting software Star-Search. To ensure robustness, the analysis was blinded and the person counting the spots did not know the identity of the samples. For each construct, *fsSox2* foci within at least 150 cells were counted in biological duplicate.

## Code availability

All the analysis in this paper was carried out using a custom package developed for the experiment called oligoGames. The package is currently hosted on GitHub—https://github.com/cshukla/oligoGames.

## Data availability

All analyzed sequence data have been deposited in NCBI GEO under accession GSE98828.

Expanded View for this article is available online.

## Acknowledgements

The authors would like to thank Doug Richardson and Sven Terclavers at Harvard Center for Biological Imaging (HCBI) for assistance with imaging and the Bauer Sequencing Facility at Harvard University for assistance with sequencing. CJS would like to acknowledge Alejandro Reyes for advice on writing the manuscript and analyzing the data. The authors would like to thank everyone in the Rinn and Irizarry laboratories for their advice and insightful comments throughout this work. This work was supported by NIH grants R01GM083084 and R01HG005220 to RAI as well as NIH grants U01DA040612-01 and P01GM099117 to JLR. PGM was supported by the "Deutsche Forschungsgemeinschaft (DFG)" (MA5028/1-3 and MA5028/1-1).

## Author contributions

JLR conceived the project. PGM, ALM, and CG designed and carried out experiments. CJS, KDK, and RAI conceived and implemented the statistical method to detect Differential Regions. CJS and RAI designed and carried out informatic analyses. MNC designed the oligo-pool and DMS assisted with the experiments. JLR, CJS, and PGM wrote the manuscript. RAI and JLR supervised and funded the project.

## Conflict of interest

The authors declare that they have no conflict of interest.

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
