## [Review Process File · The EMBO Journal]

High-throughput identification of RNA nuclear enrichment sequences

Chinmay J Shukla, Alexandra L McCorkindale, Chiara Gerhardinger, Keegan D Korthauer, Moran N Cabili, David M Shechner, Rafael A Irizarry, Philipp G Maass & John L Rinn

Review timeline:

Submission date:	19 October 2017
Editorial Decision:	20 November 2017
Revision received:	3 December 2017
Editorial Decision:	14 December 2017
Revision received:	18 December 2017
Accepted:	20 December 2017

Editor: Anne Nielsen

Transaction Report:

1st Editorial Decision

20 November 2017

Thank you for submitting your manuscript for consideration as a resource paper by The EMBO Journal. It has now been seen by two referees whose comments are shown below.

As you will see from the reports, both referees highlight the technical quality of the work and express interest in the findings reported in your manuscript. I want to add that I had an additional round of discussion with the referees (in light of the concerns from ref #2 about the level of mechanistic insight provided), emphasizing that we are considering this study for our methods/resource section. The outcome is that both referees find the method itself and the data provided to be important and valuable to the field and support its publication here.

Given the referees' positive recommendations, I would thus like to invite you to submit a revised version of the manuscript, addressing the comments of both reviewers. I should add that it is EMBO Journal policy to allow only a single round of revision, and acceptance of your manuscript will therefore depend on the completeness of your responses in this revised version.

For the revised manuscript I would particularly ask you to focus your efforts on the following points:

- Referee #1 asks for additional clarification of the retention effects for mRNAs vs lncRNAs. While this may not give a clear either-or answer, the analysis would be informative and can presumably be done with the current dataset.
- Referee #2 asks for an additional control to test if a difference in nuclear and cytoplasmic decay rates could contribute to the localization of the reporter RNA
- We noticed that the manuscript currently has 4 main figures while a lot of data is found in the 8 supplemental figures. In order to make the data more directly accessible to the readers I'd suggest that you move some of this supplemental data into the main figures.

Thank you for the opportunity to consider your work for publication. I look forward to your revision.

 REFEREE REPORTS

Referee #1:

The study uses barcoded oligonucleotides of 110nt to identify 109 nuclear-enriched RNA regions, termed "differential regions" (DRs). Presence of DRs in exons is shown to correlate with their increased nuclear retention. The identification and validation of DR motifs is convincing, and my main request is to corroborate their role in the retention of lncRNAs vs mRNAs.

The study could address more thoroughly the question why lncRNAs are more retained in the nucleus than mRNAs. The sequence motifs enriched in the DRs are used in Fig 3H-I to show that nuclear retention of DR-containing endogenous RNAs is increased. However, all classes of RNAs are grouped for this analysis. If DRs are more common in the classes that are generally more retained for other reasons, the result could be of correlative nature. The authors would need to analyse nuclear/cytoplasmic ratios separately for different classes of transcripts, especially separating lncRNAs from mRNAs in Fig 3H-I. The authors could then compare nuclear retention of DR-containing lncRNAs vs mRNA, and DR-lacking lncRNAs vs mRNA. How does nuclear retention of lncRNAs lacking DRs compare to mRNAs containing DRs?

Other comments:

- P-values are reported in the figures as 2.2×10^{-16} and for the same test in the text as $1/10^6$. Why are these values different?
- The statistics should be reported for the motifs in F3A,D: for a discriminative analysis, the number of motifs in the DR and in the non-DR sets, and the associated p-value for each motif should be reported.
- The Kolmogorov-Smirnov test might be more appropriate than the Mann-Whitney test in F3G-I, as it is usually used as a goodness-of-fit test to check if a set of continuous values follows a certain theoretical distribution, since this test relies on the statistical distance between the cumulative distribution probabilities as opposed to testing the actual values.
- The authors report that some of the DRs overlap with repC in Xist, which is known to be involved in the nuclear localisation. As I recall, repC is quite long, and I cannot see any DRs around this region. It would help to show the overlap with repC and repD in supplementary material.

Referee #2:

In this manuscript Shukla et al. describe the use of a massive reporter assay to study the ability of RNA motifs to affect nuclear localization. In particular the authors use a tiling strategy to cover the regions of 38 well-characterized lncRNAs. They perform nuclear fractionation and identify RNA motifs associated to nuclear enrichment. Using MEME the authors identify a region commonly repeated in XIST, and a C-rich region present in multiple lncRNA. Finally the authors validate by smFISH that the identified XIST sequence, and other long differential regions associated to known lncRNAs (but not the C-rich motif) contribute to the nuclear localization. I found the proposed MPRNA approach very powerful and creative. However, I feel that the current study is still mainly descriptive and lacks the mechanistic insight that I would expect for EMBO Journal.

Major concerns:

1. Although the authors do not explain why they use a frame-shifted Sox1, I would imagine it is for the potential effect of NMD if an artificial stop codon is introduced in the library. Can the author determine if the observed change of localization is caused by variations in nuclear/cytoplasmic degradation? For example comparing the abundance of reporter DNA vs the RNA expression of the same reporters and their relative localization?

Minor:

1. Some figures are redundant. For example Fig 1D seems significant by definition (as it was based on the criteria to select the DR)

2. In Fig2F. What is the distribution of the tested regions (not only the differential) regarding the FISH classes? That information is necessary to know the background distribution and to be able to measure the enrichment.

1st Revision - authors' response

3 December 2017

We thank the reviewers for their positive and constructive comments of our manuscript entitled “High-throughput identification of RNA nuclear enrichment sequences.” We have taken this opportunity to significantly improve the quality of our manuscript based on the reviewers’ suggestions. To that end, we have revised the manuscript as described in the following point-by-point response to their comments.

Referee #1:

The study uses barcoded oligonucleotides of 110nt to identify 109 nuclear-enriched RNA regions, termed "differential regions" (DRs). Presence of DRs in exons is shown to correlate with their increased nuclear retention. The identification and validation of DR motifs is convincing, and my main request is to corroborate their role in the retention of lncRNAs vs mRNAs.

We appreciate the helpful comments of the reviewer and the careful revision of our manuscript. We have addressed the reviewer’s concerns by performing new analyses, and by adding new results to the revised manuscript. Please find below a point-by-point response to the specific comments.

The study could address more thoroughly the question why lncRNAs are more retained in the nucleus than mRNAs. The sequence motifs enriched in the DRs are used in Fig 3H-I to show that nuclear retention of DR-containing endogenous RNAs is increased. However, all classes of RNAs are grouped for this analysis. If DRs are more common in the classes that are generally more retained for other reasons, the result could be of correlative nature. The authors would need to analyze nuclear/cytoplasmic ratios separately for different classes of transcripts, especially separating lncRNAs from mRNAs in Fig 3H-I. The authors could then compare nuclear retention of DR-containing lncRNAs vs mRNA, and DR-lacking lncRNAs vs mRNA. How does nuclear retention of lncRNAs lacking DRs compare to mRNAs containing DRs?

We agree, and thank the reviewer for suggesting the neglected comparison of lncRNA vs mRNA nuclear retention. As suggested, we reanalyzed the nuclear enrichment of the identified motifs within the DRs by RNA classes. We found that the identified motifs have a stronger effect on nuclear retention of lncRNAs compared to mRNAs, but their presence still significantly increases the nuclear retention of both mRNAs and lncRNAs. We modified the results section accordingly, and replaced the CDF plots in figure 3 (panels H-I) with box plots (shown below), summarizing the results of the new analysis.

Figure 3: ... H–I. Novel nuclear enrichment motifs influence the localization of endogenous human transcripts. Comparison of the nuclear enrichment of both human mRNA and lncRNA transcripts with at least one occurrence of the discovered motifs, relative to all other transcripts, in HeLa and A549 cells³⁰. P-value: Mann Whitney Test.

Other comments:

- P-values are reported in the figures as 2.2×10^{-16} and for the same test in the text as $1/10^6$. Why are these values different?

We thank the reviewer for noting this discrepancy. The correct p-value is $< 1/10^6$ as found by the Wilcoxon test. We have now corrected this typo in the figures.

- The statistics should be reported for the motifs in F3A, D: for a discriminative analysis, the number of motifs in the DR and in the non-DR sets, and the associated p-value for each motif should be reported.

We thank the reviewer for these suggestions and have now included the exact E value for each motif reported by MEME.

- The Kolmogorov-Smirnov test might be more appropriate than the Mann-Whitney test in F3G-I, as it is usually used as a goodness-of-fit test to check if a set of continuous values follows a certain theoretical distribution, since this test relies on the statistical distance between the cumulative distribution probabilities as opposed to testing the actual values.

Based on the reviewer's comment we did perform the KS test that showed a statistically significant nuclear enrichment of the motifs similar to the Mann-Whitney test. The KS test was performed to test if there was a significant difference between nuclear enrichment of lncRNAs and mRNAs harboring the motifs compared to those without the motifs and gave a p-value $< 1/10^6$. Since both Mann-Whitney and KS tests are non-parametric and both show the significance of nuclear enrichment of motifs, we report the Mann-Whitney p-value, which, in our opinion, is a more appropriate test for evaluating significant shift of the medians. However, since we originally reported the shift in the median between all transcripts (lncRNAs and mRNAs) with and without the motif, we have now reanalyzed the enrichment data independently for the 2 class of transcripts, as suggested by the reviewer in point 1 and generated box plots that are more appropriate than the previous CDF plots (see above – Figure 3H and 3I).

- The authors report that some of the DRs overlap with repC in Xist, which is known to be involved in the nuclear localization. As I recall, repC is quite long, and I cannot see any DRs around this region. It would help to show the overlap with repC and repD in supplementary material.

We agree with the reviewer that showing the overlap would facilitate the interpretation of this finding. Thus, we have now included a schematic visualization of the location of the identified XIST DRs relative to all XIST repeats in the appendix (Appendix_Figure_S1, Panel A, see below).

Figure S1. Sequence Features of Differential Regions A. UCSC genome browser tracks showing the overlap of XIST repeats and DRs.

Referee#2:

In this manuscript Shukla et al. describe the use of a massive reporter assay to study the ability of RNA motifs to affect nuclear localization. In particular the authors use a tiling strategy to cover the regions of 38 well-characterized lncRNAs. They perform nuclear fractionation and identify RNA motifs associated to nuclear enrichment. Using MEME the authors identify a region commonly repeated in XIST, and a C-rich region present in multiple lncRNA. Finally, the authors validate by smFISH that the identified XIST sequence, and other long differential regions associated to known lncRNAs (but not the C-rich motif) contribute to the nuclear localization. I found the proposed MPRNA approach very powerful and creative. However, I feel that the current study is still mainly descriptive and lacks the mechanistic insight that I would expect for EMBO Journal.

We thank the reviewer for acknowledging that “our MPRNA technique is powerful and the described results are novel and interesting for the fields of RNA biology” and for the helpful insights. We have addressed the reviewer’s concerns by performing new analysis, and by adding new results to the revised manuscript. Please find below a point-by-point response.

Major concerns:

1. Although the authors do not explain why they use a frame-shifted Sox1, I would imagine it is for the potential effect of NMD if an artificial stop codon is introduced in the library. Can the author determine if the observed change of localization is caused by variations in nuclear/cytoplasmic degradation? For example, comparing the abundance of reporter DNA vs the RNA expression of the same reporters and their relative localization?

We apologize for not describing the *fsSox2* reporter in detail. The *fsSox2* reporter was used and optimized in a previous study from our lab (Hacisuleyman et al. 2015). We previously determined that if the frameshift is not present, the transcript accumulates around the ER during translation which could influence cellular localization (moreover cell biology with induced Sox2 protein). Due to the frameshift in the *fsSox2* reporter, we were able to avoid translation-derived artefacts. While assessing nuclear/cytoplasmic degradation would not be feasible, our FISH analyses, both here and in Hacisuleyman et al. (2015), showed consistent cytosolic expression levels of *fsSox2* across different cell lines and various tested constructs, indicating uniform *fsSox2* processing. Altogether, our observations of uniform expression levels by smRNA-FISH show that (i) *fsSox2* is cytoplasmic and consistent in expression and (ii), this pattern changes with appended nuclear localization sequences (e.g. *MALATI*). This data suggests that *fsSox2* is a reasonable reporter for this proof-of principle. However, we agree with the reviewer that depending on the assay, other reporters may need to be optimized or characterized. We have now included a description why we used *fsSox2* in the text.

Minor:

1. Some figures are redundant. For example, Fig 1D seems significant by definition (as it was based on the criteria to select the DR)

We thank the reviewer for this suggestion. We now removed Fig 1D.

2. In Fig2F. What is the distribution of the tested regions (not only the differential) regarding the FISH classes? That information is necessary to know the background distribution and to be able to measure the enrichment.

The reviewer raises an interesting point about the distributions of transcripts in which DRs were called. We think the reviewer is considering a scenario where the length of transcripts in FISH class 1 were longer and would thus have more DRs by length and not by multiplicity. To address this, we now report the number of DR sequences/kb of sequence tiled, which would help us to remove any effect of different background sequences. We indeed see that this normalization removes the differences in DRs found in different FISH classes since all the FISH classes seem to harbor ~10 DRs per 10 kb of sequence tested (see Figure 2F).

Figure 2: ... F. The number of DRs discovered per 10 kb of lncRNA sequence tiled, is highly similar for each FISH Class.

2nd Editorial Decision

14 December 2017

Thank you for submitting a revised version of your manuscript. It has now been seen by one of the original referees and this person's comments are shown below. As you will see the referee finds that all criticisms have been sufficiently addressed and recommend the manuscript for publication. However, before we can go on to officially accept the manuscript, I have to ask you to address the following editorial issues concerning text and figures [...].

Thank you again for giving us the chance to consider your manuscript for The EMBO Journal, I look forward to receiving your final revision.

 REFEREE REPORT

Referee #1:

The authors have adequately addressed all of my suggestions, and I find the manuscript ready for publication, and of great interest to the broad readership of the EMBO Journal.

Corresponding Author Name: John Rinn

Journal Submitted to: EMBO

Manuscript Number: EMBOJ-2017-98452R